# Bis(triphenylphosphine)iminium Salts of Dioxothiadiazole Radical Anions: Preparation, Crystal Structures, and Magnetic Properties

Paweł Pakulski[ID], Mirosław Arczyński * and Dawid Pinkowicz *[ID]

Jagiellonian University, Faculty of Chemistry, Gronostajowa 2, 30-387 Kraków, Poland; pawel.pakulski@doctoral.uj.edu.pl

\* Correspondence: miroslaw.arczynski@doctoral.uj.edu.pl (M.A.); dawid.pinkowicz@uj.edu.pl (D.P.)

**Abstract:** Phenanthroline dioxothiadiazoles are redox active molecules that form stable radical anions suitable for the construction of supramolecular magnetic materials. Herein, the preparation, structures and magnetic properties of bis(triphenylphosphine)iminium (PPN) salts of [1,2,5] thiadiazole[3,4-f][1,10]phenanthroline 1,1-dioxide (**L**), [1,2,5]thiadiazole[3,4-f][4,7]phenanthroline 1,1-dioxide (**4,7-L**), 5-bromo-[1,2,5]thiadiazolo[3,4-f][1,10]phenanthroline 2,2-dioxide (**BrL**), and 5,10-dibromo-[1,2,5]thiadiazolo[3,4-f][1,10]phenanthroline 2,2-dioxide (**diBrL**) are reported. The preparation of new bromo derivatives of the **L**: 5-bromo-[1,2,5]thiadiazolo[3,4-f][1,10]phenanthroline 2,2-dioxide (**BrL**) and 5,10-dibromo-[1,2,5]thiadiazolo[3,4-f][1,10]phenanthroline 2,2-dioxide (**diBrL**)—suitable starting materials for further derivatization—are described starting from a commercially available and cheap 1,10-phenanthroline. All PPN salts show antiferromagnetic interactions between the pairs of radical anions, which in the case of **PPN(diBrL)** are very strong ($-116$ cm$^{-1}$; using $\hat{H} = -2JSS$ type of exchange coupling Hamiltonian) due to a different crystal packing of the anion radicals as compared to **PPN(L)**, **PPN(4,7-L)**, and **PPN(BrL)**.

**Keywords:** radical anion; redox; magnetism; antiferromagnetic coupling; dioxothiadiazole

## 1. Introduction

Purely organic molecular materials that show electric conductivity and non-trivial magnetic properties [1–4] are at the forefront of molecular materials science due to the tremendous flexibility and tunability of organic molecules. Moreover, the potential synergy and interplay between the properties of redox-active organic molecules and metal complexes open new routes to redox-active multistable systems [5,6], single molecule magnets (SMMs) and single chain magnets (SCMs) [7–9], and switchable magnetic conductors [10,11]. Achievement of such advanced properties requires, however, an expansion of the library of easily accessible and electroactive molecules with relatively stable radical forms. The most commonly studied radicals comprise α-diimines [12], dithiolenes [13–15], oxolenes [16,17], nitronyl nitroxides [18,19], tetrathiafulvalene (TTF) derivatives [20], π-conjugated macrocycles [21], TCNE and TCNQ derivatives [22–25], and verdazyl radicals [26].

Dioxothiadiazole-based electroactive molecules [27–33] and their complexes [34], on the other hand, are still underrepresented, despite their obvious advantages: easy reduction to a stable radical form, good coordination abilities (including bridging mode) and chemical tunability presented here for the first time. In 2011, Awaga et al. studied [1,2,5]thiadiazole[3,4-f][1,10]phenanthroline 1,1-dioxide (**L**) in context of its electrochemical properties and synthesis of radical salts [35]. Later on, the same group reported a number of radical salts of **L**, which revealed efficient π-orbitals overlap [36,37] transmitting efficient magnetic exchange interactions varying in strength from ferro- to very strong antiferromagnetic.

Some of us focused on the design of a dioxothiadiazole derivative with the ability to bridge two 3d metals [34]: [1,2,5]thiadiazole[3,4-f][4,7]phenanthroline 1,1-dioxide (**4,7-L**). The bridging potential was tested resulting in the formation of a coordination chain ([CuCl$_2$(**4,7-L**)]}$_n$. Both **L** and **4,7-L** reveal very similar mild reduction potentials to the radical form

Introduction of substituents to tune the redox potential of organic molecules is a well-known concept in electrochemistry [26,38]. We used the same approach to shift the reduction potentials of phenanthroline based dioxothiadiazole derivatives towards less negative values. Herein, we present two new members of the dioxothiadiazole family: 5-bromo-[1,2,5]thiadiazolo[3,4-f][1,10] phenanthroline 2,2-dioxide (**BrL**) and 5,10-dibromo-[1,2,5]thiadiazolo[3,4-f][1,10]phenanthroline 2,2-dioxide (**diBrL**) (Figure 1) and expand the library of potential spin carrying substrates by preparing and investigating bis(triphenylphosphine)iminium (PPN) salts of **L**, **4,7-L**, **BrL** and **diBrL** anion-radicals. The synthetic scheme is presented concisely in Figure 1. Structurally, these compounds reveal alternating cation–anion layers with the exception of the **PPN(diBrL)** which comprises chains of π-conjugated molecules. The influence of the structural differences on the magnetic properties of the organic salts is analyzed.

**Figure 1.** Schematic view of the synthetic route to **BrL**, **diBrL** and their radical salts: **PPN(BrL)**, **PPN(diBrL)** (**a**) and (**b**) structural formulas of **PPN(L)** and **PPN(4,7-L)**. The reaction conditions and yields are as follows: (1) $H_2SO_{4(conc.)}$, KBr, 3 h reflux, yield 95%; (2) $H_2SO_{4(98\%)}$, $HNO_{3(99\%)}$, KBr, 16 h reflux, yield 9% for diBr-phendione and 22% for Br-phendione; (3) Ethanol$_{(anh.)}$, sulfamide (3 portions), 7 d reflux, yield 91% for **BrL**, 35% for **diBrL**; (4) anhydrous acetonitrile, NaI, sonication; (5) THF, PPNCl, yield 11% for **PPN(BrL)**, and 22% for **PPN(diBrL)**.

## 2. Materials and Methods

Chemicals and reagents were of analytical grade unless otherwise stated. Inert PureSolv-MD-5/7 solvent purification system with alumina filled columns and argon gas was used for the deoxygenation and dehydration of acetonitrile and tetrahydrofuran. 1,10-phenanthroline-5,6-dione (**phendione**) was prepared according to literature procedure [39]. Some of the operations that required inert gas atmosphere were performed using Inert PureLab HE glove box filled with Ar gas.

### *2.1. Syntheses*

2.1.1. 3-Bromo-1,10-phenanthroline-5,6-dione (**Br-phendione**) and 3,8-Dibromo-1,10-phenanthroline-5,6-dione (**diBr-phendione**)

Bromo derivatives of phendione were prepared according to the modified literature procedure [40]. 1,10-phenanthroline-5,6-dione (**phendione**) (20 g, 95.2 mmol) was dissolved in a chilled mixture of 120 mL 98% $H_2SO_4$ and 60 mL 99% $HNO_3$ in a 500 mL round-bottom flask. KBr (20 g, 168 mmol) was added and the mixture was refluxed at 120 °C overnight. After 16 h, the mixture was allowed to cool down to room temperature and poured onto 500 mL of crushed ice, pH was carefully adjusted to around 5 using 0.25 M NaOH solution. The mixture was extracted with chloroform (6 × 150 mL) and the combined organic extracts were dried using $MgSO_4$ and evaporated to dryness. Column chromatography ($SiO_2$, eluent: $CHCl_3$:AcOEt:n-hexane, 10:2:1) afforded pure 3-bromo-1,10-phenanthroline-5,6-dione (6.0 g, 22%), 3,8-dibromo-1,10-phenanthroline-5,6-dione (3.0 g, 9%) and the unreacted phendione (0.9 g). 3-bromo-1,10-phenanthroline-5,6-dione [1]H NMR ($CDCl_3$) δ [ppm]: 9.14 (d, 1H, *J* = 2.5 Hz), 9.11 (dd, 1H, *J* = 4.8, 1.9 Hz), 8.60 (d, 1H, *J* = 2.5 Hz), 8.50 (dd, 1H, *J* = 7.8, 1.8 Hz), 7.61 (dd, 1H, *J* = 7.9, 4.7 Hz); 3,8-dibromo-1,10-phenanthroline-5,6-dione [1]H NMR ($CDCl_3$) δ [ppm]: 9.15 (d, 2H, *J* = 2.3 Hz), 8.62 (d, 2H, *J* = 2.5 Hz). [1]H NMR spectra can be found in the Supplementary Materials.

2.1.2. 5-Bromo-[1,2,5]thiadiazolo[3,4-f][1,10]phenanthroline 2,2-Dioxide (**BrL**)

3-bromo-1,10-phenanthroline-5,6-dione (1.0 g, 3.5 mmol) was suspended in 35 mL of anhydrous ethanol, sulfamide (0.56 g, 5.8 mmol) was added and the mixture refluxed for seven days. After 24 h, another quantity (0.25 g) of sulfamide was added. Additions were repeated daily for one week. Next, the mixture was allowed to cool down to room temperature and filtered. The yellow solid was washed with two aliquots of ethanol (2 × 20 mL) and dried in vacuo to afford 1.1 g (91%) of the desired product. [1]H NMR ($CDCl_3$) δ [ppm]: 9.22 (d, 1H, *J* = 2.5 Hz), 9.19 (dd, 1H, *J* = 4.8, 1.8 Hz), 8.84 (d, 1H, *J* = 2.3 Hz), 8.72 (dd, 1H, J = 7.9, 1.8 Hz), 7.67 (dd, 1H, *J* = 7.9, 4.7 Hz). [1]H NMR spectra can be found in the Supplementary Materials.

2.1.3. 5,10-Dibromo-[1,2,5]thiadiazolo[3,4-f][1,10]phenanthroline 2,2-dioxide (**diBrL**)

3,8-dibromo-1,10-phenanthroline-5,6-dione (150 mg, 0.4 mmol) was suspended in 20 mL of anhydrous ethanol, sulfamide (50 mg, 0.5 mmol) was added and the mixture refluxed for three days. After 24 h, another quantity (50 mg) of sulfamide was added. Additions were repeated daily for the duration of the synthesis. Next, the brownish-orange mixture was allowed to cool down to room temperature and filtered to afford a brownish-yellow which was washed with two aliquots of ethanol (2 × 10 mL) and dried under vacuum. The crude product was once more suspended in 20 mL of the ethanolic solution of sulfamide (50 mg) and refluxed for another 24 h. The mixture was allowed to cool down to room temperature and filtered. An orange solid was washed with 20 mL of ethanol, 10 mL of cold diethyl ether and dried under vacuum to afford 198 mg (17%) of the desired product. [1]H NMR ($CDCl_3$) δ [ppm]: 9.19 (d, 2H, *J* = 2.2 Hz), 8.42 (d, 2H, *J* = 2.3 Hz). [1]H NMR spectra can be found in the Supplementary Materials.

2.1.4. Bis(triphenylphosphine)iminium [1,2,5]thiadiazole[3,4-f][1,10]phenanthroline 1,1-Dioxide $H_2O/(CH_3)_2CO$ Solvate (**PPN(L)**)

This compound was obtained in a two-step procedure: the suspension of 310 mg of [1,2,5]thiadiazole[3,4-f][1,10]phenanthroline 1,1-dioxide (1.15 mmol) (**L**) in 90 mL of MeCN was stirred overnight with 4 g of NaI (26.7 mmol) under ambient atmosphere. The resulting dark red precipitate was filtered and washed with 60 mL of MeCN until the colour of the product **Na(L)** changed from dark red to purple. The **Na(L)** was dried in vacuo for a few hours. Anal. calcd. for $C_{12}H_7N_4NaO_{2.5}S$ (Na(1,10-tdapO$_2$)·0.5 H$_2$O): C, 47.68; H, 2.33; N, 18.54; S, 10.61. Found: C, 47.75; H, 2.45; N, 18.55; S, 10.20.

293 mg of **Na(L)**·0.5H$_2$O was dissolved in 210 mL of acetone followed by the addition of 2.0 g (3.6 mmol) of PPNCl and stirring for 40 min. After that time the purple/violet solution was filtered to remove the precipitated NaCl. The filtrate was concentrated in a rotary evaporator to ca. 10 mL, which was left undisturbed for 1 h for crystallization.

The product was collected by decantation, filtered and washed with a single drop of cold acetone. Product was dried in air. Yield 660 mg (75%) based on **Na(L)**. Anal. calcd. for **PPN(L)**·$H_2O$·$(CH_3)_2CO$, $C_{51}H_{44}N_5O_4P_2S$ (884.9 g/mol): C, 69.22; H, 5.01; N, 7.91; S, 3.62. Found: C, 68.97; H, 4.89; N, 7.96; S, 4.06.

2.1.5. Bis(triphenylphosphine)iminium [1,2,5]thiadiazole[3,4-f][4,7]phenanthroline 1,1-Dioxide (**PPN(4,7-L)**)

Product was obtained similarly to **PPN(L)**. The suspension of 63 mg (0.23 mmol) of [1,2,5]thiadiazole[3,4-f][4,7]phenanthroline 1,1-dioxide (**4,7-L**) in 35 mL of dry MeCN was stirred overnight under an inert atmosphere with a 1.0 g (6.6 mmol) of NaI. Next day the dark violet precipitate was separated from the mother solution by filtration and washed with dry MeCN (ca. 20 mL). The powder attains violet colour after washing. After vacuum drying for a few hours the 30 mg of crude **Na(4,7-L)** was used to obtain **PPN(4,7-L)**. This was achieved by stirring MeCN (135 mL) suspension with 210 mg of PPNCl for two hours. The NaCl precipitate was filtered off and a clear violet solution was quickly condensed on a rotary evaporator to ca. 6 mL and transferred to the Ar-filled glovebox, where it was left for 2 h for crystallization. Large elongated block crystals were separated from the mother suspension by decantation, filtered and washed with a single drop of cold MeCN. Yield 49 mg (59% based on **Na(4,7-L)**). Anal. calcd. for **PPN(4,7-L)**, $C_{48}H_{36}N_5O_2P_2S$ (808.8 g/mol): C, 71.28; H, 4.4; N, 8.66; S, 3.96. Found: C, 70.67; H, 4.30; N, 8.62; S, 4.12.

2.1.6. Bis(triphenylphosphine)iminium 5-Bromo-[1,2,5]thiadiazole[3,4-f][1,10]phenanthroline 1,1-dioxide THF Solvate (**PPN(BrL)**)

350 mg of **BrL** (1.0 mmol) and 2.0 g of NaI (13.3 mmol) was sonicated for 20 min in anhydrous acetonitrile (50 mL). During sonication the yellow suspension turned dark violet due to the reduction of **BrL** to a radical anion $BrL^{•−}$ by iodide. The mixture was filtered and the dark violet precipitate was washed with three portions of anhydrous acetonitrile (3 mL each). The sodium salt **Na(BrL)** was dried under vacuum for 1 h (300 mg, 81%) and then suspended in dry tetrahydrofuran (50 mL). Solid PPNCl (1.15 g, 2.0 mmol) was added resulting in the color change of the liquid phase to deep violet and the precipitation of white sodium chloride. NaCl was removed by filtration and the violet THF solution was concentrated by rotary evaporation to ca. 15 mL. Large crystals of the THF solvate were obtained by slow vapor diffusion of dry diethyl ether onto the THF mother solution (two days). The violet crystals (ca. 1–3 mm) were separated by hand under the microscope from the smaller colorless crystals of PPNCl. Yield 90 mg (11% based on **BrL**). Anal. calcd. for **PPN(BrL)**·2THF, $C_{56}H_{51}BrN_5O_4P_2S$ (1032.0 g/mol): C: 65.18, H: 4.98, 6.79, S: 3.11. Found: C: 65.49, H: 5.15, N: 6.58, S: 2.71.

2.1.7. Bis(triphenylphosphine)iminium 5,10-Dibromo-[1,2,5]thiadiazole[3,4-f][1,10]phenanthroline 1,1-Dioxide (**PPN(diBrL)**)

150 mg of **diBrL** (0.35 mmol) and 1.0 g of NaI (6.6 mmol) was sonicated for 10 min in anhydrous acetonitrile (20 mL). During the sonication, the yellow suspension turned dark violet due to the reduction of **diBrL** to a radical anion by iodide. The mixture was filtered and the dark violet precipitate was washed with three portions of anhydrous acetonitrile (2 mL each). The sodium salt **Na(diBrL)** was dried under vacuum for 1 h (95 mg, 60%) and then suspended in dry tetrahydrofuran (50 mL). Solid PPNCl (0.4 g, 0.7 mmol) was added in small portions resulting in the color change of the liquid phase to deep violet and the precipitation of sodium chloride. NaCl was removed by filtration and the violet THF solution was evaporated to dryness. The crude **PPN(diBrL)** was dissolved in ca. 15 mL of MeCN. Then the solution was evaporated to ca. 2 mL. The sample was left for an hour for crystallization and then the crystals were separated and purified similar to **PPN(4,7-L)**. Yield 45 mg (22%). Anal. calcd. for **PPN(diBrL)**, $C_{48}H_{34}Br_2N_5O_2P_2S$ (966.6 g/mol): C: 59.64, H: 3.55, N: 7.25, S: 3.32. Found: C: 59.32, H: 3.46, N: 7.15, S: 3.05.

*2.2. Other Physical Measurements*

IR spectra were collected using Nicolet iN10 MX FT-IR microscope in the transmission mode. Cyclic voltammetry was performed using Mtm-anko M-161C electrochemical analyzer. Glassy carbon electrodes were used in both experiments. [1]H NMR spectra were measured using Bruker Avance II 300 MHz spectrometer. Elemental CHNS analysis were done with ELEMENTAR Vario Micro Cube CHNS analyzer.

2.2.1. Magnetic Measurements

The magnetic measurements were carried out using Quantum Design MPMS3 Evercool SQUID magnetometer. The samples were sealed in HDPE foil bags to protect them from the crystallization solvent loss. Corrections for the diamagnetism of the sample holder and the compounds themselves (Pascal constants) were applied [41].

2.2.2. X-ray Diffraction Data Collection/Refinement

Single crystal X-ray diffraction (XRD) data was collected on a Bruker D8 Quest Eco diffractometer equipped with Photon 50[TM] CMOS detector and Mo-K$_\alpha$ Triumph® monochromator. The data were collected at low temperature using Kryoflex II low-temperature device. Data were integrated using SAINT [42], while multi-scan absorption corrections were applied using SADABS or TWINABS [43,44], all incorporated into APEX3 environment [45]. The structures were solved using SHELXT [46] and refined with SHELXL [47,48] software within the Olex2 package [49]. All hydrogen atoms were refined using riding model, and the non-hydrogen atoms were refined anisotropically using weighted full-matrix least-squares on F$^2$. Disorder of THF crystalline solvent molecules in **PPN(BrL)** was refined using constraints. Ideal THF geometry was supported by IMGL library [50]. The occupancy factors of THF molecules are changed so that the refinement could be stable. The positions should not be taken as perfect because the model shows just one possibility. One of the phenyl rings of PPN cation is also strongly disordered, apparently over four positions, but it was refined using only two of them, where positive electron densities were the strongest. **PPN(diBrL)** was refined as two component twin with scales 0.6227(6). In case of **PPN(L)** the occupation factor of the water oxygen atom was fixed at low value due to disorder of the solvent molecules.

CCDC 1882326-1882331 contains the supplementary crystallographic data for this paper. These data can be obtained free of charge via http://www.ccdc.cam.ac.uk/conts/retrieving.html (or from the CCDC, 12 Union Road, Cambridge CB2 1EZ, UK; Fax: +44 1223 336033; E-mail: deposit@ccdc.cam.ac.uk).

### 2.2.3. Calculation Details

Density functional theory (DFT) calculations were done in Gaussian09 program [51]. For each molecule hybrid Becke 3-parameter Lee-Yang-Parr (B3LYP) [52,53] functional with 6-311++G(2d,2p) basis set was used [54]. Single crystal XRD structural models were taken as a starting geometry. No structural constraints were imposed on any atom.

## 3. Results

### 3.1. Syntheses

The **Br-phendione** and **diBr-phendione** were synthesized by bromination of **phendione** using KBr in the mixture of concentrated $HNO_3$ and $H_2SO_4$ acids (Figure 1). The products were purified by column chromatography. Both were further used in the procedure of attaching the thiadiazole dioxide functional group to the phenanthroline backbone. This includes the reaction of alpha-diketone group with sulfamide in the anhydrous boiling ethanol (Figure 1). In contrast to the preparation of **L** or **4,7-L**, reactions of sulfamide with **Br-phendione** and **diBr-phendione** require longer time and additional quantities of sulfamide added every 24 h of the reaction under reflux to achieve good yields. The synthesis of all PPN radical salts was carried out in two steps. In the first step, the reduction of the respective dioxothiadiazole derivative with sodium iodide led to the quantitative precipitation of poorly soluble sodium salts. In the second step, the metathesis of the obtained sodium salts using PPNCl resulted in the final PPN salts, which are very well soluble in THF, acetonitrile, dichloromethane, and chloroform (Figure 1).

### 3.2. BrL and diBrL—Crystal Structures and DFT Calculations

Crystal structures of **BrL** and **diBrL** were determined using single crystal X-ray diffraction (Table S1 in the Supplementary Materials). **BrL** crystallizes in the *P*-1 space group while **diBrL** in $P2_1/c$. The asymmetric units of both compounds are presented in Figure 2a,b. **BrL** and **diBrL** molecules are equipped with bromine substituents which change the crystal packing of the molecules as compared to the non-substituted **L**. **BrL** forms π-π stacks along "*a*" crystallographic direction with the parallel off-centered arrangement of the molecules forced by steric hindrance of the $SO_2$ group (Figure S1a in the Supplementary Materials). The plane of each molecule in a stack is inclined relative to the direction of stack propagation. The structure of **BrL** seems to be similar to that of **L** with the layers of molecules forming a two-dimensional (2-D) network of double N···H-C hydrogen bonds (donor acceptor distance of 3.525 Å on one side of the molecule and 3.642 Å on the other). The presence of the Br substituent in **BrL** disrupts the H-bonding and the molecular packing as compared to **L**. One side of the molecule forms slightly different interactions, namely N···S close contacts (3.331 Å) which replace the N···H-C bonds (Figure S1b in the Supplementary Materials). Also, the donor acceptor distances in N···H-C hydrogen bonds of **BrL** are a little shorter (3.467 Å) compared to those in **L**. The molecules of **BrL** do not lie completely flat, but are slightly tilted directing the bromine atoms slightly below the plane of the neighboring molecule.

The structure of the **diBrL** is completely different. The chains of parallel hydrogen bonded dimers interacting with each other through π-orbitals and bromine atoms are easily distinguished (Figure S2 in the Supplementary Materials). The neighboring chains run in two perpendicular directions, and the source of closest contacts between them are N and O atoms of the dioxothiadiazole groups.

The DFT calculations (B3LYP exchange-correlation functional; 6-311++G(2d,2p) basis set) were performed for **BrL** and **diBrL**. Noteworthy, the inspection of molecular orbitals revealed that low-lying LUMO is not spread over the bromine atoms in contrast to the HOMO (Figure 3). The LUMOs for both **BrL** and **diBrL** molecules exhibit an antibonding character at the N-C bonds of the dioxothiadiazole group (nodal plane) and a bonding character at the dioxothiadiazole's S-N and C-C bonds, which is consistent with the observed shortening and elongation of the respective bonds discussed below.

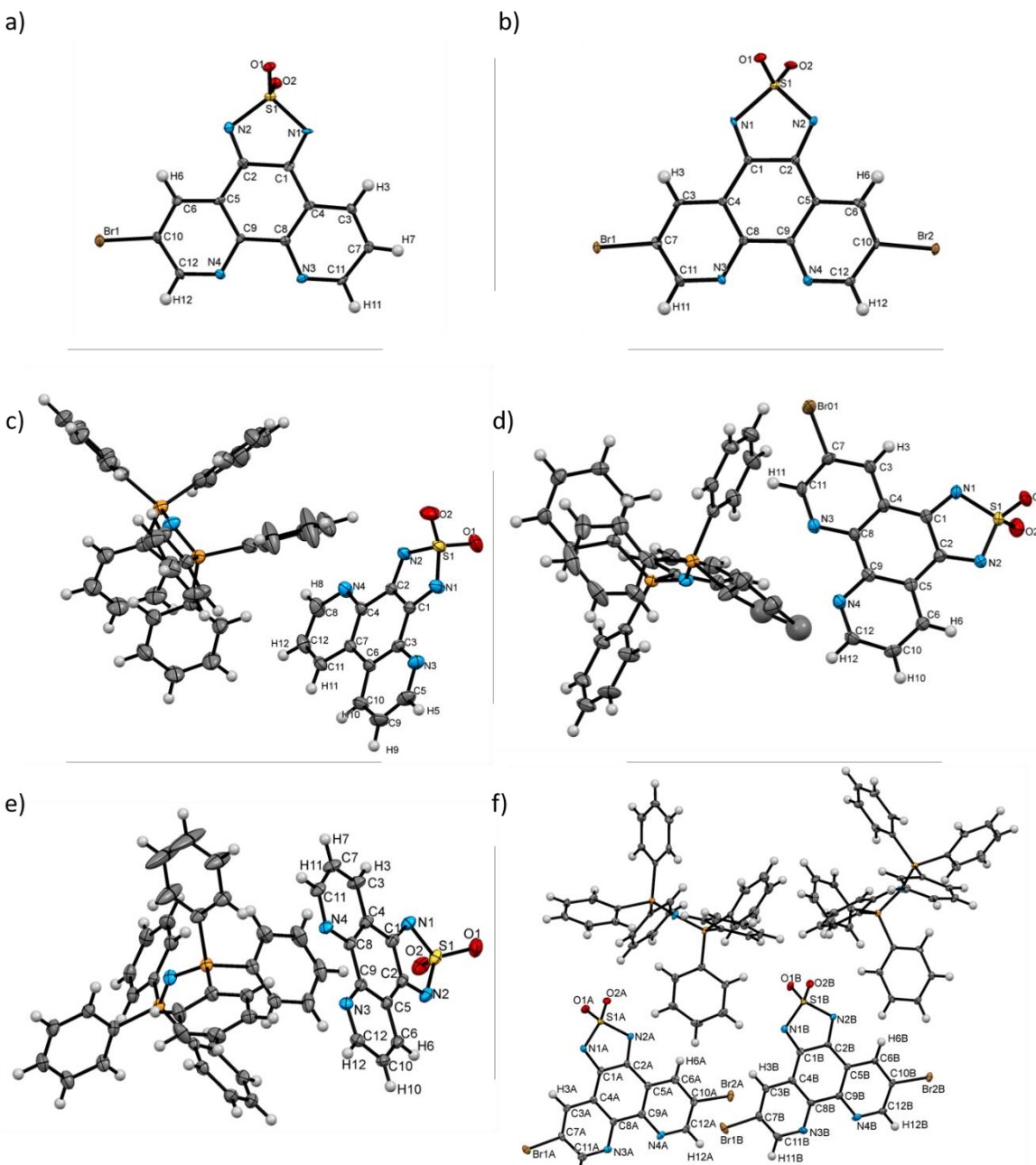

**Figure 2.** Asymmetric units of (**a**) **BrL**; (**b**) **diBrL**; (**c**) **PPN(4,7-L)**; (**d**) **PPN(BrL)**; (**e**) **PPN(L)**; (**f**) **PPN(diBrL)**; crystallization solvent molecules and disorder treated parts omitted for the sake of clarity.

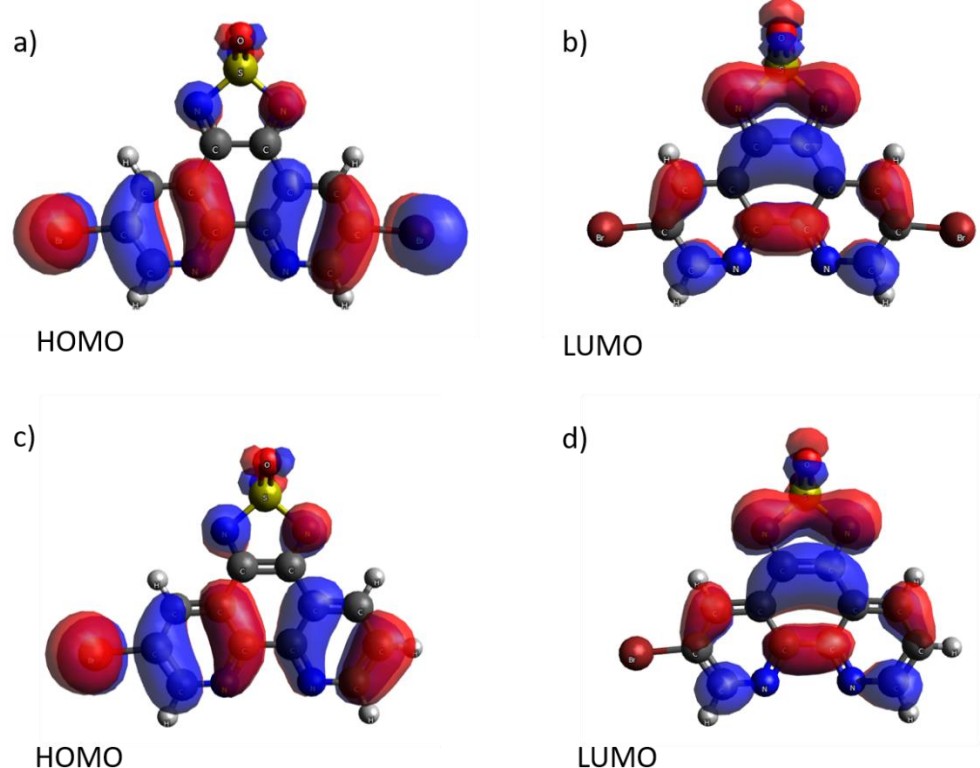

**Figure 3.** Results of the DFT B3LYP calculations results: HOMO of **diBrL** (**a**), LUMO of **diBrL** (**b**), HOMO of **BrL** (**c**), and LUMO of **BrL** (**d**).

### 3.3. BrL and diBrL—Cyclic Voltammetry

Electrochemical properties of brominated **BrL** and **diBrL** are similar to the previously reported **L** [35] and **4,7-L** [34] featuring two distinct reduction processes (Figure 4). The first reduction leads to an anion radical and appears at ca. −441 mV vs Fc⁺/Fc. The second one results in a diamagnetic dianion and appears ca. 800 mV below the first reduction. The exact values of the reduction potentials are presented in Table 1. The first reduction potential of **diBrL** (−441 mV) and **BrL** (−471 mV) are shifted to less negative values as compared to **L** (−499 mV) and **4,7-L** (−520 mV). This trend is also reflected in the HOMO-LUMO gap, which is much smaller for the brominated species (Table 1).

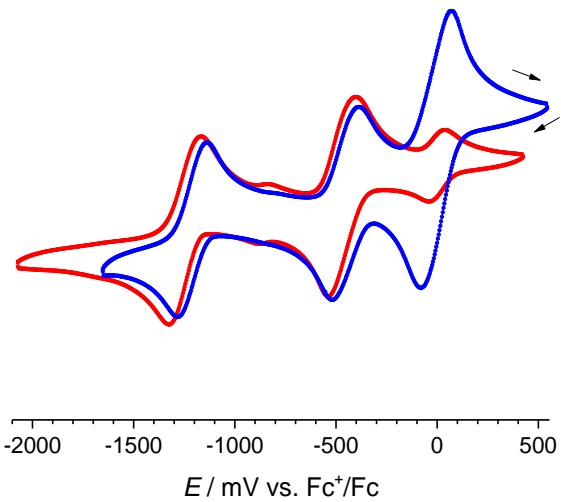

$E$ / mV vs. Fc⁺/Fc

**Figure 4.** Cyclic voltammograms of **BrL** (red line) and **diBrL** (blue line) in 100 mM MeCN solution of n-Bu4NPF6 recorded at 100 mV/s vs. Fc⁺/Fc.

**Table 1.** Reduction half-potentials and HOMO-LUMO gap (Δ) of the dioxothiadiazole-based molecules.

| Compound | Reduction Wave No. | $E_{ox}$/mV | $E_{red}$/mV | $E_{1/2}$/mV | Δ/eV | Ref. |
|----------|--------------------|--------------|---------------|----------------|-------|------|
| **diBrL** | 1 | −387 | −495 | −441 | 3.183 | this work |
|           | 2 | −1139 | −1279 | −1208 |       |           |
| **BrL**  | 1 | −405 | −538 | −471 | 3.352 | this work |
|          | 2 | −1166 | −1324 | −1245 |       |           |
| **L**    | 1 |      |      | −499 | 3.562 | [35] |
|          | 2 |      |      | −1320 |       |       |
| **4,7-L** | 1 |      |      | −520 | 3.583 | [34] |
|           | 2 |      |      | −1229 |       |       |

### 3.4. PPN Radical Salts—Crystal Structures

Structural X-ray diffraction data revealed that **PPN(L)**, **PPN(BrL)**, and **PPN(diBrL)** crystallize in *P*-1 while PPN(**4,7-L**) crystallizes in *P*2₁/c space group (Table S1 in the Supplementary Materials).

The asymmetric unit (ASU) of **PPN(diBrL)** contains two anion-radicals and two PPN⁺ cations while ASUs of **PPN(4,7-L)**, **PPN(L)**, and **PPN(BrL)** comprise only one respective radical anion and one counter-cation (Figure 2). **PPN(4,7-L)** and **PPN(diBrL)** crystallize without solvent molecules while **PPN(L)** incorporates one acetone and one water molecule and **PPN(BrL)** crystallizes with tetrahydrofuran molecules.

In terms of crystal packing **PPN(L)**, **PPN(BrL)**, and **PPN(4,7-L)** exhibit alternating anion-cation layered arrangement presented in Figure 5. Due to this particular arrangement the radical anions in these three organic salts exhibit negligible π-orbital overlap and a number of N···H-C hydrogen bonds (Figure 5) that connect neighboring anions in pairs as indicated in Figure 5 by the green ovals. Crystal packing of **PPN(diBrL)** is completely different and comprises infinite π-π stacks of radical anions separated by PPN cations (Figure 5d). The shortest D···A (H-bond) or π-π contacts connecting the dioxothiadiazole groups, where the spin density is the highest, is as follows: 3.403 Å (**PPN(4,7-L)**), 3.520 Å (**PPN(BrL)**), 3.455 Å (**PPN(L)**) and 3.322 Å (**PPN(diBrL)**). Please note, that in the case of (**PPN(4,7-L)**) the separation of dioxothiadiazole groups is the most efficient, despite the shortest D···A distances.

As already mentioned, **L**$^{\bullet-}$ and **BrL**$^{\bullet-}$ anion radicals in their respective PPN salts form almost flat anionic layers (Figure 5b,c) while **4,7-L**$^{\bullet-}$ layer is significantly corrugated (as in corrugated cardboard) (Figure 5a).

Similarly to previously reported dioxothiadiazoles, a one electron reduction results in the contraction of the C-C and S-N bonds (by ca. 0.06 Å and 0.04 Å, respectively), as well as the elongation of C=N and S=O (by ca. 0.05 Å and 0.015 Å, respectively). These results line up with the shape of the LUMO, which is occupied in the radical form [34,35,37]. These particular bond lengths are indicative of the oxidation state of the molecule. Table 2 compares bond lengths of neutral and radical dioxothiadiazole-based compounds presented here.

As can be seen from Table 2, the bond lengths in all presented compounds do not deviate from those of previously reported dioxothiadiazoles. Most notable difference between neutral and anion-radical molecules are found in C-C bond lengths which are shortened due to one electron reduction by more than 0.06 Å as compared to the neutral form (from ca 1.51 Å to ca. 1.44 Å, respectively).

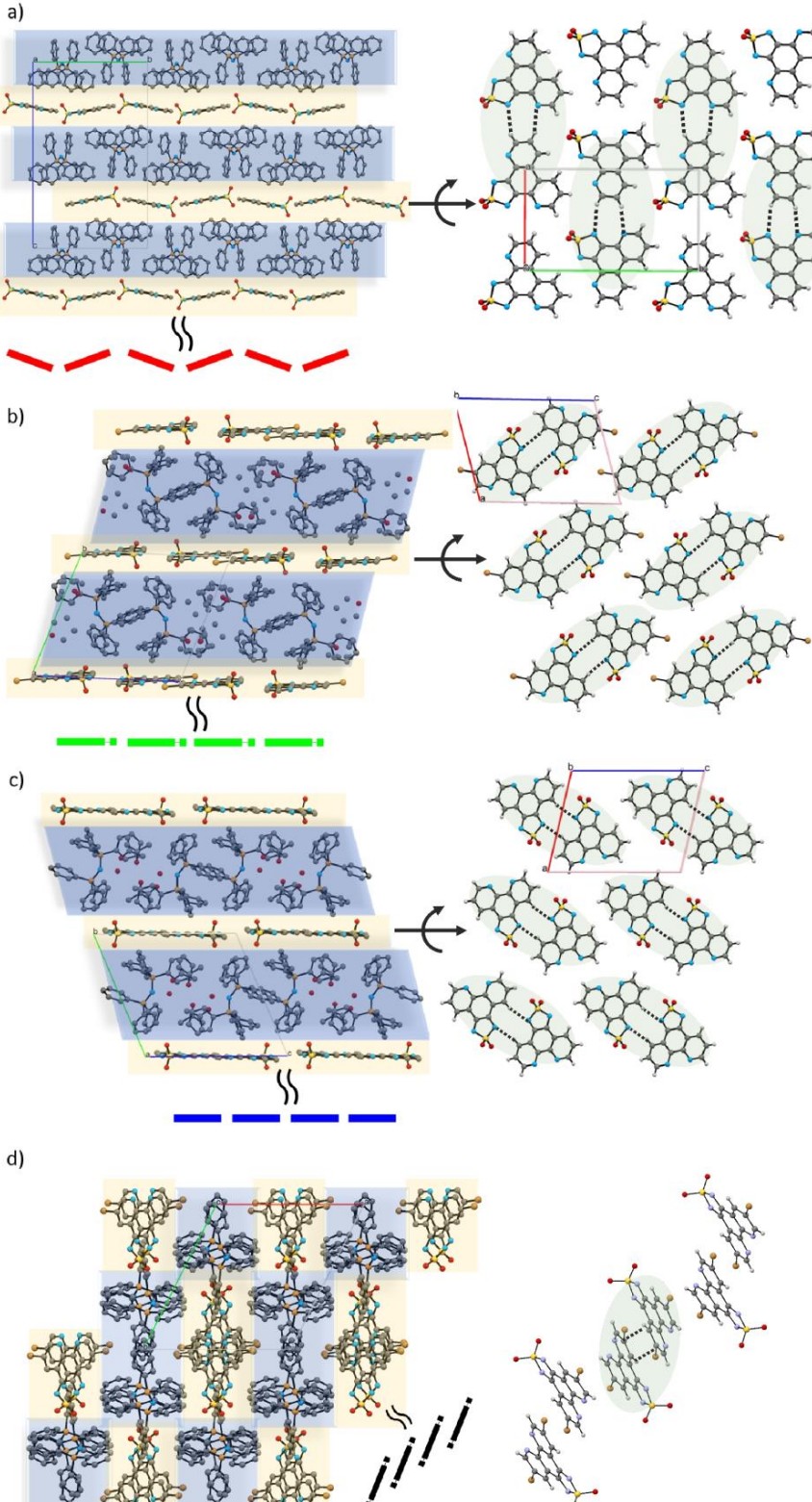

**Figure 5.** Packing diagrams of the layered structural models of (**a**) **PPN(4,7-L)**; (**b**) **PPN(BrL)**; (**c**) **PPN(L)**; and (**d**) **PPN(diBrL)**; Cationic PPN$^+$ layers are marked with blue colour and anionic radical layers are marked with yellow for clarity. On the right side is the top view of a single radical anion layer. Note that the anionic layers of **PPN(4,7-L)** as well as **PPN(BrL)** are not completely flat, as the radical anions are slightly tilted. The green ovals indicate the shortest D···A (H-bond) or $\pi$-$\pi$ contacts connecting the dioxothiadiazole groups: 3.403 Å (**a**), 3.520 Å (**b**), 3.455 Å (**c**), and 3.322 Å (**d**).

**Table 2.** Selected bond lengths (in Å) of BrL, diBrL vs. PPN(4,7-L), PPN(BrL), PPN(L), PPN(diBrL).

| Compound | S=O | S-N | C=N | C-C |
|---|---|---|---|---|
| **Br-L** | 1.423(4)<br>1.429(4) | 1.693(5)<br>1.695(4) | 1.284(7)<br>1.293(6) | 1.505(7) |
| **diBr-L** | 1.421(4)<br>1.426(4) | 1.693(4)<br>1.696(5) | 1.289(7)<br>1.284(7) | 1.514(7) |
| **av. in neutral molecules** | **1.425** | **1.694** | **1.288** | **1.510** |
| **PPN(4,7-L•)** | 1.439(2)<br>1.436(2) | 1.648(2)<br>1.657(2) | 1.333(2)<br>1.333(2) | 1.443(2) |
| **PPN(Br-L•)** | 1.437(3)<br>1.433(3) | 1.649(3)<br>1.660(3) | 1.333(5)<br>1.332(5) | 1.441(5) |
| **PPN(1,10-L•)** | 1.442(2)<br>1.448(2) | 1.664(2)<br>1.664(3) | 1.342(3)<br>1.340(3) | 1.443(4) |
| **PPN(diBr-L•) Molecule B** | 1.443(4)<br>1.443(4) | 1.646(5)<br>1.660(5) | 1.336(7)<br>1.333(8) | 1.452(8) |
| **PPN(diBr-L•) Molecule A** | 1.442(4)<br>1.444(4) | 1.656(5)<br>1.660(5) | 1.338(8)<br>1.333(8) | 1.435(8) |
| **av. in radical anions** | **1.441** | **1.656** | **1.335** | **1.443** |

### 3.5. PPN$^+$ Radical Salts—Magnetic Properties

The results of magnetic measurements are presented in both Figure 6 and Table 3. In all three salts that reveal layered structures (**PPN(4,7-L)**, **PPN(L)**, and **PPN(BrL)**) the $\chi T(T)$ curves show very similar behavior. This dependence is constant above ca. 60 K and takes the values which are close to the theoretical 0.375 cm$^3$ K mol$^{-1}$ spin-only value assuming $S = \frac{1}{2}$ and $g = 2.0$ (Table 3 and Figure 6). Below ca. 60 K the $\chi T(T)$ decreases and plummets below 15 K achieving values close to zero due to antiferromagnetic interactions between the radical anions. $M(H)$ curves at 1.8 K differ slightly among these three compounds. The $M(H)$ are slowly, almost linearly increasing with field (Figure 6b), until at some point the increase becomes steep, which again is typical for weak local antiferromagnetic interactions between neighboring spins. For **PPN(4,7-L)** the inflection point is located around 4 T, while for **PPN(L)**—around 6 T and for **PPN(BrL)** well above 7 T. The $M(H)$ dependencies do not saturate at 7 T and the magnetization values at this field decreases along the series **PPN(4,7-L)**, **PPN(BrL)**, **PPN(L)**, suggesting that the strongest magnetic interactions operate within the **PPN(L)** salt.

**Table 3.** Values of $\chi T$ at 300 K and 1.8 K, M(H) at 7T and the magnetic exchange constants $J$ obtained from fitting of $M(H)$ and $\chi T(T)$ in the whole temperature range using PHI software [55].

| Compound | $\chi T(T)$ @300K/cm$^3$ K mol$^{-1}$ | $\chi T(T)$ @80K/cm$^3$ K mol$^{-1}$ | $\chi T(T)$ @1.8K/cm$^3$ K mol$^{-1}$ | $M(H)$ @7T/$\mu_B$ | $J$/cm$^{-1}$ |
|---|---|---|---|---|---|
| **PPN(4,7-L)** | 0.374 | 0.377 | 0.075 | 0.76 | −2(1) |
| **PPN(BrL)** | 0.374 | 0.363 | 0.024 | 0.20 | −4(1) |
| **PPN(L)** | 0.367 | 0.352 | 0.006 | 0.07 | −5(1) |
| **PPN(diBrL)** | 0.313 * | 0.200 | 0.157 ** | 0.37 | −116(10)<br>−0.6(5) |

* at 340 K; ** at 7 K.

$\chi T(T)$ curve for **PPN(diBrL)** (Figure 6 black dots), on the other hand, decreases in the whole 340–80 K temperature range from 0.313 cm$^3$ K mol$^{-1}$ at 340 K to 0.196 cm$^3$ K mol$^{-1}$ at 80 K with a plateau-like feature around 0.186 cm$^3$ K mol$^{-1}$ below this temperature. The signal starts to decrease again below 30 K and reaches a minimum of 0.158 cm$^3$ K mol$^{-1}$ at 7 K. Near 2.0 K additional small increase of the $\chi T(T)$ signal is observed which might be ascribed to very weak ferromagnetic interactions between the radical anions. The $M(H)$ curve increases in a Brillouin like fashion reaching the value of 0.37 $\mu_B$ (well below the expected 1.0 $\mu_B$ for $S = \frac{1}{2}$ and $g = 2.0$), but close to 0.5 $\mu_B$ which

suggests the presence of strong antiferromagnetic interactions between half of the radical anions in the compound.

The magnetic data were fitted assuming local antiferromagnetic interactions between the pairs of anion radicals which in the case of **PPN(4,7-L)**, **PPN(L)**, and **PPN(BrL)** are transmitted through C-H···N hydrogen bonds (Figure 5) and for **PPN(diBrL)** through the π-π contacts with different interplane distances (two types of radical pairs with two types of magnetic interactions (one very strong and the other very weak). Figure S8 presents the magnetic coupling scheme for all four compounds. The results of the simultaneous fitting of $\chi T(T)$ and $M(H)$ (PHI program [55]) using the following Hamiltonians (Equation (1) for **PPN(4,7-L)**, **PPN(L)**, and **PPN(BrL)** and Equation (2) for **PPN(diBrL)**) are collected in Table 3 and presented as solid lines in Figure 6

$$\hat{H} = -2J_{12} \cdot S_1 \cdot S_2 + \mu_B \cdot g_1 \cdot S_1 \cdot B + \mu_B \cdot g_2 \cdot S_2 \cdot B \tag{1}$$

$$\hat{H} = -2J_{12} \cdot S_1 \cdot S_2 + -2J_{34} \cdot S_3 \cdot S_4 + \mu_B \cdot g_1 \cdot S_1 \cdot B + \mu_B \cdot g_2 \cdot S_2 \cdot B + \mu_B \cdot g_3 \cdot S_3 \cdot B + \mu_B \cdot g_4 \cdot S_4 \cdot B \tag{2}$$

where $S_1 = S_2 = S_3 = S_4 = \frac{1}{2}$ are the spin numbers of the radical anions, $g_1 = g_2 = g_3 = g_4 = 2.0$ is the $g$-factor, $\mu_B$ is the Bohr magneton, $B$ is the magnetic field induction and $J_{12}$ and $J_{34}$ are the superexchange coupling constants—the fitting parameters with $J_{12} \gg J_{34}$.

The magnetic interaction pathways between pairs of radical anions are justified by the presence of hydrogen bonded supramolecular pairs highlighted in Figure 5b,c. In this simplified model, each radical anion interacts with only one neighbor utilizing two C-H···N hydrogen bonds. These hydrogen bonds are the strongest mediators of magnetic interactions in the structures of **PPN(4,7-L)**, **PPN(L)**, and **PPN(BrL)** justifying the use of single exchange coupling parameter. In case of **PPN(diBrL)** four spin carriers were taken into account operating with two different exchange coupling constants $J_{12}$ and $J_{34}$, assuming that $J_{12} \gg J_{34}$. This is dictated by the presence of two step-like features in the $\chi T(T)$ dependence.

The $\chi T(T)$ fits correspond well with the experimental data above 10 K and the antiferromagnetic exchange coupling increases with the increasing temperature at which the decrease of the $\chi T$ occurs. The weak match between the fitted and experimental $M(H)$ is a consequence of a simplified model employed in the analysis of the magnetic data and the presence of non-interacting $S = 1/2$ spins due to the defects in the crystal structure. However, the inflection of the $M(H)$ curves at 4 T for **PPN(4,7-L)**, 6 T for **PPN(L)**, and >7 T for **PPN(BrL)** is followed by the increase of the antiferromagnetic exchange coupling in this series.

The **PPN(4,7-L)**, **PPN(BrL)**, and **PPN(L)** belong to a structurally-related series where the anions and cations are arranged in layers. It appears that the magnetic interactions are strongly related to this arrangement. The strongest interactions are achieved in completely flat layers of **PPN(L)** with $J_{12} = -5(1)$ cm$^{-1}$. In **PPN(BrL)** the bulky bromine substituent increases the separation between the radicals and disrupts the C-H···hydrogen bonds resulting in slightly weaker magnetic interactions ($J_{12} = -4(1)$ cm$^{-1}$). Finally, in **PPN(4,7-L)** the layer is composed of tilted molecules with much weaker C-H···N H-bonds and the estimated magnetic interactions are even weaker ($J_{12} = -2(1)$ cm$^{-1}$). While this magneto-structural correlation is simplified, it clearly demonstrates how the derivatization of dioxothiadiazole-based radical anions enables fine-tuning of their magnetic behavior.

Magnetic behavior of **PPN(diBrL)** is also strongly correlated with the structural packing. The stacks of **diBr-L$^{\bullet-}$** radical anions reveal four different π-contacts between them leading to two types of radical anion pairs within the infinite stack (Figure S9 in the Supplementary Materials). The most efficient one controls the magnetic behavior of one half of the radical anions and results in very strong antiferromagnetic coupling ($J_{12} = -116(10)$ cm$^{-1}$) that is comparable with the values reported for sodium salts of **4,7-L** [34] and **L** [35] and other types of molecular magnets ([56] and references therein). The antiferromagnetic interactions result in a $\chi T$ value of 0.313 cm$^3$ K mol$^{-1}$ at 340 K which is significantly lower than the expected 0.375 cm$^3$ K mol$^{-1}$ for non-interaction $S = 1/2$ species. These interactions lead also to a plateau of 0.186 cm$^3$ K mol$^{-1}$ below 70 K corresponding to

half of the radical anions in the compound. The second exchange parameter $J_{34}$ is much weaker than $J_{12}$ but seems to be slightly underestimated as the fit does not reproduce the second step around 30 K where a further decrease of the $\chi T(T)$ to 0.157 cm³ K mol⁻¹ occurs. This weaker exchange controls the magnetic behavior of the remaining half of radical anions and is responsible for the observation of the 0.186 cm³ K mol⁻¹ plateau below 70 K and the Brillouin-like $M(H)$ curve reaching a saturation value approaching 0.5 $\mu_B$ expected for half of the radicals in **PPN(diBrL)** (Figure 6b).

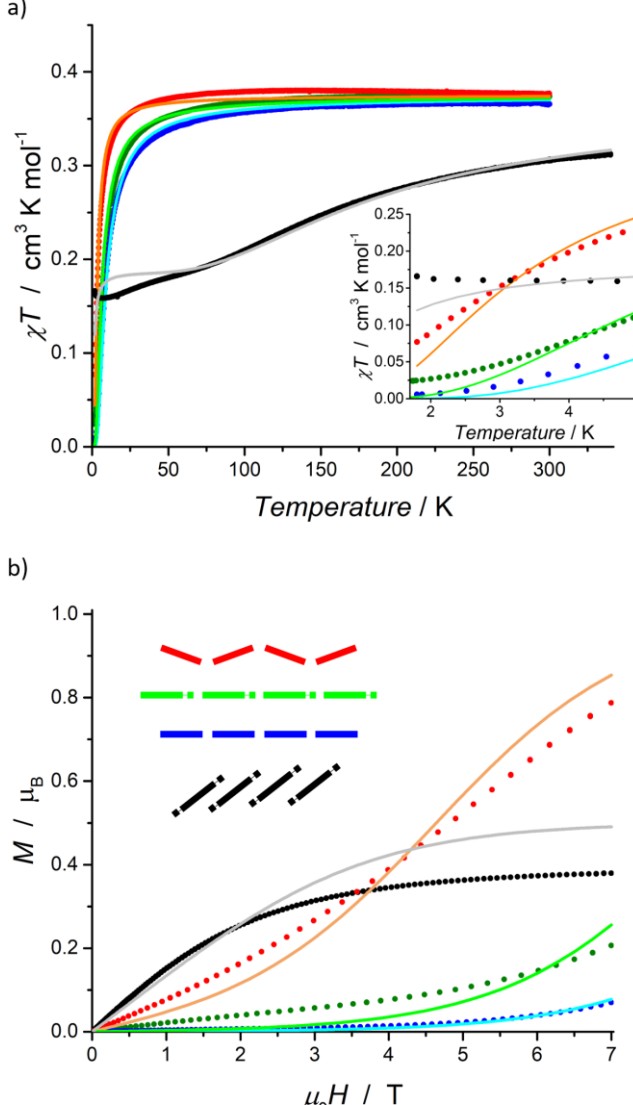

**Figure 6.** Experimental magnetic data (points) and best fits (solid lines) for **PPN(4,7-L)** (red), **PPN(L)** (blue), **PPN(BrL)** (green), and **PPN(diBrL)** (black): $\chi T(T)$ recorded at 0.1 T with an inset showing the low temperature window (**a**) and $M(H)$ recorded at 1.8 K with the schematic representation of the packing of radical anions (**b**). The $\chi T$ and $M$ values are calculated per one mole of radical anions.

## 4. Conclusions

Two new derivatives of [1,2,5]thiadiazole[3,4-f][1,10]phenanthroline 1,1-dioxide (**L**)—a redox active dioxothiadiazole—have been prepared and synthesized starting from 1,10-phenathroline: 5-bromo-[1,2,5]thiadiazolo[3,4-f][1,10]phenanthroline 2,2-dioxide (**BrL**), and 5,10-dibromo-thiadiazolo[3,4-f][1,10]phenanthroline 2,2-dioxide (**diBrL**). In the next step their organic paramagnetic salts with PPN⁺ counter-cations have been prepared along with the previously unknown PPN⁺ salts of **4,7-L** [34] and **L** [35]. The PPN salts show very good solubility in THF, acetonitrile, chloroform, and

dichloromethane, which renders them suitable for the preparation of mixed-spin systems. All four radical anion-based compounds were characterized by means of single-crystal X-ray diffraction and magnetic measurements (SQUID magnetometry). **PPN(L)**, **PPN(4,7-L),** and **PPN(BrL)** exhibit layered-type structures where the flat/weaved anionic layers are separated by layers of PPN$^+$ cations. **PPN(diBrL)**, on the other hand, forms infinite chain-like π-π stacks of radical anions that are separated from each other by cations. The structures of the reported compounds directly influence the magnetic properties. The 'layered salts' show weak-to-moderate antiferromagnetic interactions despite slightly different substituents (bromine atoms) or the location of the nitrogen atoms, while the 'π-π-stacked-salt' exhibits very strong antiferromagnetic interactions transmitted through the direct overlap of the π orbitals of the radical anions. **PPN(L)**, **PPN(4,7-L),** and **PPN(BrL)** constitute a rare example of a layered packing where the layers of radical anions are separated by the layers of cations (similar packing was observed for a few other PPN-based supramolecular systems [57]).

**Supplementary Materials:** The following are available online at http://www.mdpi.com/2073-4352/9/1/30/s1. Table S1. Details of single crystal X-ray data and structural refinement for **PPN(L)** (CCDC 1882329) **PPN(BrL)** (CCDC 1882331), **PPN(4,7-L)** (CCDC 1882328), **PPN(diBrL)** (CCDC 1882330), **diBrL** (CCDC 1882327), and **BrL** (CCDC 1882326). Figure S1. Illustration of molecular stacks in **BrL** with marked short σ-π contacts a) and fragment of a supramolecular layer with marked contacts that are shorter than the sum of the Van der Waals radii b) (CCDC Mercury program). Figure S2. Illustration of crystal packing of **diBrL**. The supramolecular chains of parallel hydrogen bonded dimers run through the structure interacting via π-orbitals and short contacts with bromine atoms. Figure S3. Illustration of **PPN(BrL)** supramolecular layers. Contacts between BrL anions that are shorter than the sum of the Van der Waals radii. Figure S4. NMR spectrum of 3-bromo-1,10-phenantroline-5,6-dione., Figure S5. NMR spectrum of 5-bromo-[1,2,5]thiadiazole[3,4-f] phenanthroline 2,2-dioxide. Figure S6. NMR spectrum of 3,8-dibromo-1,10-phenanthroline-5,6-dione. Figure S7. NMR spectrum of 5,10-dibromo-[1,2,5]thiadiazole[3,4-f][1,10]phenanthroline 2,2-dioxide. Figure S8. Superexchange coupling scheme in **PPN(4,7-L)** (a), **PPN(BrL)** (b), **PPN(L)** (c), and **PPN(diBrL)** (d). The green ovals and dotted lines indicate the magnetic interaction pathways taken into account in the fitting of the magnetic data. In the case of **PPN(diBrL)** (d) two interaction pathways are considered: $J_{12}$ and $J_{34}$ with the assumption that $J_{12} >> J_{34}$. Figure S9. Illustration of supramolecular stacks of diBrL anions in the crystal structure of **PPN(diBrL)**. The molecules are color coded to depict different intermolecular contacts between them. The asymmetric unit contains one green and one blue molecule. The most efficient π-π overlap is between the light blue and navy blue colored radical anions.

**Author Contributions:** Conceptualization: D.P., P.P., and M.A.; Methodology: P.P., M.A., and D.P.; Validation: D.P. and M.A.; Formal analysis: D.P. and M.A.; Investigation: P.P., M.A., and D.P.; Data curation: D.P. and M.A.; Writing—original draft preparation, M.A. and P.P.; Writing—review and editing: D.P.; Visualization: M.A.; Supervision: D.P.; Project administration: D.P.; Funding acquisition: M.A. and D.P.

**Funding:** This research was funded by the Polish National Science Centre within the Sonata Bis 6 (2016/22/E/ST5/00055) project. M.A. gratefully acknowledges the Polish Ministry of Science and Higher Education for the financial support within the Diamond Grant project (0041/DIA/2015/44).

**Acknowledgments:** This research was supported in part by PLGrid Infrastructure.

**Conflicts of Interest:** The authors declare no conflict of interest.

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
