# Peer review of "Bis(triphenylphosphine)iminium Salts of Dioxothiadiazole Radical Anions: Preparation, Crystal Structures, and Magnetic Properties"

_crystals, doi:10.3390/cryst9010030_

Reviewer 1 Report

This work is well done and  interesting. I therefore recommend publication. There are several typing errors (e. g. page 3, line 65: "salts" instead of "slats".

Reviewer 2 Report

The manuscript describes the synthesis, crystal structures, DFT MO calculations and magnetic properties of a series of radical anion salts prepared with phenantroline dioxothiadiazole derivatives and its mono and di bromo derivatives. The results are interesting and the amount of work is quite high. Therefore, the manuscript might be accepted for publication although there are some important points that need to be improved as follow:

1.     The synthesis of all the compounds should be described in a more easy to understand way. There are many short sentences without articles. It is a kind of telegram. Also, the volume units should be changed to mL, rather than ml, as recommended by the IUPAC.

2.     In the synthesis of (PPN)L the acronym 1,10-tdapO2 appears for the first time without any explanation.

3.     The stepped increase observed in the M vs H plots indicate the presence of a metamagnetic behavior in all cases with quite high critical fields rather than antiferromagnetic interactions as indicated in the text (lines 325 and 326).

4.     The XT value of PPN(diBrL) at 340 K is 0.313 cm3 K mol-1 in line 335 but 0.326 cm3 K mol-1 in table 3. The value in figure 6a seems to be close to 3.1 cm3 K mol-1. Please, indicate the correct value.

5.     The non-interacting defects and paramagnetic impurities should not give rise to an increase in XT at low temperatures for PPN(diBrL). In this compound the M vs H plot follows a Brillouin function but with a very small contribution. This fact, and the plateau reached at low temperatures in the XT plot suggests that this sample presents a very high residual paramagnetic contribution (almost ½ electron). This is a very strange contribution and must be explained by the authors.

6.     In general, the fitting of the magnetic properties, specially the M vs. H curves, are very poor, most probably due to the metamagnetic behavior of compounds PPN(4,7-L), PPN(1,10-L) and PPN(Br-L). Authors should confirm this metamagnetic behavior by measuring (and displaying) the magnetic susceptibility vs T at different DC applied fields below and above the critical fields (although in PPN(1,10-L) and PPN(Br-L) the critical field seems to be above 7 T). The low values of the magnetic coupling agrees with the possible presence of a metamagnetic behavior.

7.     Other minor errors are: “slats” in line 65. In the elemental analysis results of PPN(diBrL) there is an N lacking in line 172. Space groups should be written with the letters in italics (not the numbers).

Reviewer 3 Report

The paper reports on the synthesis, crystal structures and magnetic characterizations of a series of dioxothiadiazole radical compounds. The structures and magnetic properties of the prepared compounds are well characterized and discussed. It is therefore a pleasure for me to recommend the manuscript for publication. However, before doing so I request the authors to address the following minor revisions.

1.      I recommend to revise the abbreviations of PPN(BrL), PPN(diBrL), PPN(L) and PPN(4,7-L) to more appropriate ones with a contrasting scheme. I think that the abbreviations of compound 1, 2, 3 and 4 are fully satisfied.

2.      P. 12, L342-344. The authors mentioned that the magnetic data of the compounds PPN(4,7-L), PPN(L) and PPN(BrL) were fitted assuming local antiferromagnetic interactions through the pairs of anion radicals transmitting by C-H…N hydrogen bonds without any description about C-H…N hydrogen bonding dimeric radical anion in the crystal structural part. In the crystal structural part, there has only 2D network of PPN(BrL) through C-H…N hydrogen bonding mentioned (P. 6, L236-237), which is different to the assumption by authors in the magnetic part.

3.      To added a magnetic coupling scheme of compounds PPN(BrL), PPN(diBrL), PPN(L) and PPN(4,7-L) was suggested.

4.      Figure 6a, the fitting result of compound PPN(4,7-L) is not match to the experimental data. It is suggested to fit the experimental data ignored the data in the low-temperature range.

5.      Figure 6b, the magnetization data of the compounds PPN(BrL), PPN(diBrL), PPN(L) and PPN(4,7-L) showed a various behavior. To compare the magnetization data with magnetic coupling and discus was suggested.

After corrections of these points I will be happy to see the paper published in Crystals.

Author Response

Response attached

Round  2

Reviewer 2 Report

Authors have answered to all my questions but there is still one open question. I still believe that the compounds may behave as metamagnets with a very high critical field (this is supported by the weak magnetic coupling and the fact that the peaks in X(T) are not so rounded, specially when plotted in a 0-300 K scale as are usually represented). Any case, the best way to be sure is to measure the X(T) plots in the low T region with very high applied DC magnetic fields (up to 7 T). If the maximum disappears for very high magnetic fields then we can deduce that compounds are metamagnets if not, I will agree with the authors.
